complexity/applied mathematics/graph theory

directed graph, spectral methods, network model, community structure, graph embedding, graph Laplacian

**Author for correspondence:**
Xue Gong
e-mail: X.Gong-8@sms.ed.ac.uk

# Directed network Laplacians and random graph models

Xue Gong[1,2], Desmond J. Higham[1] and

Konstantinos Zygalakis[1]

[1]School of Mathematics, University of Edinburgh, Edinburgh EH9 3FD, UK
[2]The Maxwell Institute for Mathematical Sciences, Edinburgh EH8 9BT, UK

 XG, 0000-0002-6470-8679; DJH, 0000-0002-6635-3461;
KZ, 0000-0002-3860-9167

We consider spectral methods that uncover hidden structures in directed networks. We establish and exploit connections between node reordering via (a) minimizing an objective function and (b) maximizing the likelihood of a random graph model. We focus on two existing spectral approaches that build and analyse Laplacian-style matrices via the minimization of frustration and trophic incoherence. These algorithms aim to reveal directed periodic and linear hierarchies, respectively. We show that reordering nodes using the two algorithms, or mapping them onto a specified lattice, is associated with new classes of directed random graph models. Using this random graph setting, we are able to compare the two algorithms on a given network and quantify which structure is more likely to be present. We illustrate the approach on synthetic and real networks, and discuss practical implementation issues.

# 1. Motivation

Uncovering structure by clustering or reordering nodes is an important and widely studied topic in network science [1,2]. The issue is especially challenging if we move from undirected to directed networks, because there is a greater variety of possible structures. For example, even a simple motif of three connected nodes has 13 distinct forms [3, fig. 1*a*]. Moreover, when spectral methods are employed, directed edges lead to asymmetric eigenproblems [4–7]. Our objective in this work is to study spectral (Laplacian-based) methods for directed networks that aim to reveal *clustered, directed, hierarchical structure*; that is, groups of nodes that are related because, when visualized appropriately, one group is seen to have links that are directed towards the next group. This hierarchy may be periodic or linear, depending on whether there are well-defined start and end groups. Figure 1*a,b* illustrates the two cases. Mapping a network to a linear structure may help us understand the upstreamness and downstreamness of nodes, which is useful, for example, in the study of cascading effects such as social or

**Figure 1.** Directed networks with (*a*) periodic hierarchy (edges point from nodes in one cluster to nodes in the next cluster, counterclockwise) and (*b*) linear hierarchy (edges point from nodes in one level to nodes in the next highest level). Node colours indicate the three clusters.

financial contagion [8]. Similarly, periodic hierarchies have been associated with sustainability and risk management issues in commerce [9], and also with the existence of echo chambers in online social media [10].

Of course, on real data, these structures may not be so pronounced; hence in addition to visualizing the reordered network, we are interested in quantifying the relative strength of each type of signal. Laplacian-based methods are often motivated from the viewpoint of optimizing an objective function. This work focuses on two such methods. Minimizing *frustration* leads to the *magnetic Laplacian* which may be used to reveal periodic hierarchy [5,11]. Minimizing *trophic incoherence* leads to what we call the *trophic Laplacian*, which may be used to reveal linear hierarchy [6]. We will exploit the idea of associating a spectral method with a generative random graph model. This in turn allows us to compare the outputs from spectral methods based on the likelihood of the associated random graph. This connection was proposed in [12] to show that the standard spectral method for undirected networks is equivalent to maximum-likelihood optimization assuming a class of range-dependent random graphs (RDRGs) introduced in [13]. The idea was further pursued in [14], where a likelihood ratio test was developed to determine whether a network with RDRG structure is more linear or periodic.

The main contributions of this work are as follows.

— We propose two new directed random graphs models. One model has the unusual property that the probability of an $i \rightarrow j$ connection is not independent of the probability of the reciprocated $j \rightarrow i$ connection.
— We establish connections between these random graph models and algorithms from [6,11] that use the magnetic Laplacian and trophic Laplacian, respectively, by showing that reordering nodes or mapping them onto a specific lattice structure using these algorithms is equivalent to maximizing the likelihood that the network is generated by the models proposed.
— We show that by calibrating a given network to both models, it is possible to quantify the relative presence of periodic and linear hierarchical structures using a likelihood ratio.
— We illustrate the approach on synthetic and real networks.

The rest of the paper is organized as follows. In the next section, we introduce the magnetic and trophic Laplacian algorithms. Section 3 defines the new classes of random directed graphs and establishes their connection to these spectral methods. Illustrative numerical results on synthetic networks are given in §4, and in §5, we show results on real networks from a range of applications areas. We finish with a brief discussion in §6.

## 2. Magnetic and trophic Laplacians

### 2.1. Notation

We consider an unweighted directed graph $G = (V, E)$ with node set $V$ and edge set $E$, with no self-loops. The adjacency matrix $A$ is $n \times n$ with $A_{ij} = 1$ if the edge $i \rightarrow j$ is in $E$, and $A_{ij} = 0$ otherwise. It is

convenient to define the symmetrized adjacency matrix $W^{(s)} = (A + A^T)/2$. The symmetrized degree matrix $D$ is diagonal with $D_{ii} = d_i$, where $d_i = \sum_j W_{ij}^{(s)}$ is the average of the in-degree and out-degree of node $i$. Later, we will consider weighted networks for which each edge $i \rightarrow j$ has associated with it a non-negative weight $w_{ij}$. In this case, we let $A_{ij} = w_{ij}$. We use $\mathbf{i}$ to denote $\sqrt{-1}$, and we write $x^H$ to denote the conjugate transpose of a vector $x \in \mathbb{C}^n$. We use $\mathcal{P}$ to denote the set of all permutation vectors, that is, all vectors in $\mathbb{R}^n$ with distinct components given by the integers $1, 2, \ldots, n$.

## 2.2. Spectral methods for directed networks

Spectral methods explore properties of graphs through the eigenvalues and eigenvectors of associated matrices [1,2,15,16]. In the undirected case, the standard graph Laplacian $L = D - A$ is widely used for clustering and reordering, along with normalized variants. The directed case has received less attention; however, several extensions of the standard Laplacian have been proposed [7]. We focus on two spectral methods for directed networks, which are discussed in the next two subsections: the magnetic Laplacian algorithm, which reveals periodic flow structures [5,11], and the trophic Laplacian algorithm, which reveals linear hierarchical structures [6]. We choose to study these two algorithms because they have an optimization formulation and, as we show in §3, may be interpreted in terms of random graph models. Here, we briefly mention two other related techniques that do not fit naturally into this framework. The Hermitian matrix method groups nodes into clusters with a strong imbalance of flow between clusters [4]. This approach constructs a skew-symmetric matrix that emphasizes net flow between pairs of nodes but ignores reciprocal edges. A spectral clustering algorithm motivated by random walks was derived in [17] leading to a graph Laplacian for directed networks that was proposed earlier in [18].

## 2.3. The magnetic Laplacian

Given a network and a vector of angles $\boldsymbol{\theta} = (\theta_1, \theta_2, \ldots, \theta_n)^T$ in $[0, 2\pi)$, we may define the corresponding *frustration*

$$\eta(\boldsymbol{\theta}) = \sum_{i,j} W_{ij}^{(s)} |e^{\mathbf{i}\theta_i} - e^{\mathbf{i}\delta_{ij}} e^{\mathbf{i}\theta_j}|^2, \tag{2.1}$$

where $\delta_{ij} = -2\pi g \alpha_{ij}$ with $g \in [0, \frac{1}{2}]$. Here, $\alpha_{ij} = 0$ if the edge between $i$ and $j$ is reciprocated, that is $A_{ij} = A_{ji} = 1$; $\alpha_{ij} = 1$ if the edge $i \rightarrow j$ is unreciprocated, that is $A_{ij} = 1$ and $A_{ji} = 0$; and $\alpha_{ij} = -1$ if the edge $j \rightarrow i$ is unreciprocated, that is $A_{ij} = 0$ and $A_{ji} = 1$. For convenience, we also set $\alpha_{ij} = 0$ if $i$ and $j$ are not connected. To understand the definition (2.1), suppose that for a given graph we wish to choose angles that produce low frustration. Each term $W_{ij}^{(s)} |e^{\mathbf{i}\theta_i} - e^{\mathbf{i}\delta_{ij}} e^{\mathbf{i}\theta_j}|^2$ in (2.1) can make a positive contribution to the frustration if $W_{ij}^{(s)} \neq 0$; that is, if $i$ and $j$ are involved in at least one edge. In this case, if there is an edge from $i$ to $j$ that is not reciprocated, then we can force this term to be zero by choosing $\theta_j = \theta_i + 2\pi g$. If the edge is reciprocated, then we can force the term to be zero by choosing $\theta_j = \theta_i$. Hence, intuitively, choosing angles to minimize the frustration can be viewed as mapping the nodes into directed clusters on the unit circle in such a way that (a) nodes in the same cluster tend to have reciprocated connections and (b) unreciprocated edges tend to point from source nodes in one cluster to target nodes in the next cluster, periodically. Setting the parameter $g = 1/k$ for some positive integer $k$ indicates that we are looking for $k$ directed clusters.

On a real network, it is unlikely that the frustration (2.1) can be reduced to zero, but it is of interest to find a set of angles that give a minimum value. This minimization problem is closely related to the *angular synchronization* problem [19,20], which estimates angles from noisy measurements of their phase differences $\theta_i - \theta_j \mod 2\pi$. Moreover, we note that for visualization purposes it makes sense to reorder the rows and columns of the adjacency matrix based on the set of angles that minimizes the frustration. We also note that in [11] the expression (2.1) for the frustration is normalized through a division by $2\sum_i d_i$. This is immaterial for our purposes, since that denominator is independent of the choice of $\boldsymbol{\theta}$.

The frustration (2.1) is connected to the magnetic Laplacian, which is defined as follows, where $A \circ B$ denotes the elementwise, or Hadamard, product between matrices of the same dimension; that is, $(A \circ B)_{ij} = A_{ij} B_{ij}$.

**Definition 2.1.** Given $g \in [0, \frac{1}{2}]$, the **magnetic Laplacian** $L^{(g)}$ [5,11] is defined as

$$L^{(g)} = D - T^{(g)} \circ W^{(s)},$$

where $T_{ij}^{(g)} = e^{i\delta_{ij}}$. Here, the *transporter matrix* $T^{(g)}$ assigns a rotation to each edge according to its direction.

It is straightforward to show that $L^{(g)}$ is a Hermitian matrix. When $g = 0$ and the graph is undirected, the magnetic Laplacian reduces to the standard graph Laplacian.

The following result, which is implicit in [5,11], shows that the frustration (2.1) may be written as a quadratic form involving the magnetic Laplacian.

**Theorem 2.2.** *Let $\psi \in \mathbb{C}^n$ be such that $\psi_j = e^{i\theta_j}$, then*

$$\psi^H L^{(g)} \psi = \tfrac{1}{2} \sum_{i,j} W_{ij}^{(s)} |e^{i\theta_i} - e^{i\delta_{ij}} e^{i\theta_j}|^2. \tag{2.2}$$

Appealing to the Rayleigh–Ritz theorem [21] the quadratic form on the left-hand side of (2.2) is minimized over all $\psi \in \mathbb{C}^n$ with $\|\psi\|_2 = 1$ by taking $\psi$ to be an eigenvector corresponding to the smallest eigenvalue of the magnetic Laplacian. Now, such an eigenvector will not generally be proportional to a vector with components of the form $\{e^{i\theta_j}\}_{j=1}^n$. However, a useful heuristic is to force this relationship in a componentwise sense; that is, to assign to each $\theta_j$ the phase angle of $\psi_j$, effectively solving a relaxed version of the desired minimization problem. This leads to algorithm 1, as used in [11].

---

**Algorithm 1.** Magnetic Laplacian algorithm.

---

**Result:** Phase angles of nodes $\theta$

**Input adjacency matrix** $A$;

**Symmetrize adjacency matrix** $W^{(s)} = (A + A^T)/2$;

**Calculate degree matrix** $D_{ii} = d_i = \sum_j W_{ij}^{(s)}$;

**Construct transporter** $T_{ij}^{(g)} = e^{i\delta_{ij}}$;

**Calculate Magnetic Laplacian** $L^{(g)} = D - T^{(g)} \circ W^{(s)}$;

**Compute eigenvectors** $\{\psi_m^{(g)}\}_{m=1}^n = Eigs(L^{(g)})$ and associated eigenvalues;

**Calculate phase angles** $\theta = \mathrm{phase}(\psi_1^{(g)})$ **using eigenvector** $\psi_1^{(g)}$ **associated with the smallest eigenvalue**;

**Reorder nodes** with $\theta_i$ or **visualize** with $(\cos(\theta_i), \sin(\theta_i))$

---

## 2.4. The trophic Laplacian

The idea of discovering a linear directed hierarchy arises in many contexts where edges represent dominance or approval, including the ranking of sports teams [22] and Web pages [23]. A particularly well-defined case is the quantification of trophic levels in food webs, where each directed edge represents a consumer–resource relationship [24–26]. We focus here on the approach in [6], where the aim is to assign a trophic level $h_i$ to each node $i$ such that along any directed edge the trophic level increases by one. This motivates the minimization of the *trophic incoherence*

$$F(\boldsymbol{h}) = \frac{\sum_{i,j} A_{ij}(h_j - h_i - 1)^2}{\sum_{i,j} A_{ij}}. \tag{2.3}$$

Denoting the total weight of node $i$ as $\omega_i = \sum_{j \in V}(A_{ji} + A_{ij})$ and the imbalance as $\chi_i = \sum_{j \in V}(A_{ji} - A_{ij})$, the trophic level vector $\boldsymbol{h} \in \mathbb{R}^n$ that minimizes the trophic incoherence solves the linear system of equations

$$\Lambda \boldsymbol{h} = \chi, \tag{2.4}$$

where $\Lambda = \mathrm{diag}(\omega) - A - A^T$, and the solution to (2.4) is unique up to a constant shift [6]. Since it employs a Laplacian-style matrix, $\Lambda$, we refer to it as the *trophic Laplacian* algorithm; see algorithm 2.

**Algorithm 2.** Trophic Laplacian algorithm.

---

**Result:** The trophic levels $h$

**Input** adjacency matrix $A$;

**Calculate the node weights** $\omega_i = \sum_j A_{ji} + \sum_j A_{ij}$;

**Calculate the node imbalances** $\chi_i = \sum_j A_{ji} - \sum_j A_{ij}$;

**Calculate the trophic Laplacian** $\Lambda = \text{diag}(\omega) - A - A^T$;

**Solve** the linear system (2.4);

**Reorder** or **visualize** nodes using $h$

---

# 3. Random graph interpretation

In this section, we associate two new random graph models with the magnetic and trophic Laplacian algorithms, using a similar approach to the work in [12]. After establishing these connections, we proceed as in [14] and propose a maximum-likelihood test to compare the two models on a given network.

## 3.1. The directed pRDRG model

Given a set of phase angles $\{\theta_i\}_{i=1}^n$, we will define a model for unweighted, directed random graphs. The model generates connections between each pair of distinct nodes $i$ and $j$ with four possible outcomes—a pair of reciprocated edges, an unreciprocated edge from $i$ to $j$, an unreciprocated edge from $j$ to $i$, or no edges—as follows:

$$\mathbf{P}(A_{ij} = 1, A_{ji} = 1) = f(\theta_i, \theta_j), \tag{3.1}$$

$$\mathbf{P}(A_{ij} = 1, A_{ji} = 0) = q(\theta_i, \theta_j), \tag{3.2}$$

$$\mathbf{P}(A_{ij} = 0, A_{ji} = 1) = l(\theta_i, \theta_j) \tag{3.3}$$

and

$$\mathbf{P}(A_{ij} = 0, A_{ji} = 0) = 1 - f(\theta_i, \theta_j) - q(\theta_i, \theta_j) - l(\theta_i, \theta_j), \tag{3.4}$$

where $f$, $q$ and $l$ are functions that define the model, and, of course, they must be chosen such that all probabilities lie between zero and one. We emphasize that this model has a feature that distinguishes it from typical random graph models, including directed Erdős–Rényi and small-world style versions [27]: the probability of the edge $i \to j$ is not independent of the probability the edge $j \to i$, in general.

We are interested here in the inverse problem where we are given a graph and a model (3.1)–(3.4), and we wish to infer the phase angles. This task arises naturally when the nodes are supplied in some arbitrary order. We will assume that the phase angles are to be assigned values from a discrete set $\{v_i\}_{i=1}^n$; that is, we must set $\theta_i = v_{p_i}$, where $p$ is a permutation vector. This setting includes the cases of (directed) clustering and reordering. For example, with $n = 12$, we could specify $v_1 = v_2 = v_3 = 0$, $v_4 = v_5 = v_6 = \pi/2$, $v_7 = v_8 = v_9 = \pi$, and $v_{10} = v_{11} = v_{12} = 3\pi/2$, in order to assign the nodes to four directed clusters of equal size. Alternatively, $v_i = (i-1)2\pi/12$ would assign the nodes to equally spaced phase angles, as shown in figure 2a, as a means to reorder the graph. The following theorem shows that solving this type of inverse problem for suitable $f$, $q$ and $l$ is equivalent to minimizing the frustration.

**Theorem 3.1.** *Suppose $\boldsymbol{\theta} \in \mathbb{R}^n$ is constrained to take values such that $\theta_i = v_{p_i}$, where $p$ is a permutation vector. Then minimizing the frustration $\eta(\boldsymbol{\theta})$ in (2.1) over all such $\boldsymbol{\theta}$ is equivalent to maximizing the likelihood that the graph came from a model of the form (3.1)–(3.4) in the case where*

$$f(\theta_i, \theta_j) = \frac{1}{Z_{ij}},$$

$$q(\theta_i, \theta_j) = \frac{1}{Z_{ij}} \exp[\gamma(1 - 2\cos\beta_{ij} + \cos(\beta_{ij} + 2\pi g))]$$

$$and \quad l(\theta_i, \theta_j) = \frac{1}{Z_{ij}} \exp[\gamma(1 - 2\cos\beta_{ij} + \cos(\beta_{ij} - 2\pi g))],$$

*with $\beta_{ij} = \theta_i - \theta_j$ and normalization constant*

$$Z_{ij} = 1 + e^{\gamma(1 - 2\cos\beta_{ij} + \cos(\beta_{ij} + 2\pi g))} + e^{\gamma(1 - 2\cos\beta_{ij} + \cos(\beta_{ij} - 2\pi g))} + e^{\gamma(2 - 2\cos\beta_{ij})},$$

*for any positive constant $\gamma$.*

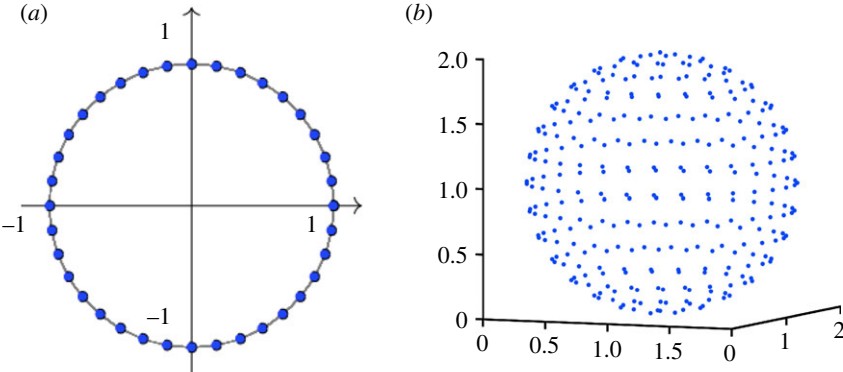

**Figure 2.** (a) Points uniformly distributed on the unit circle and (b) a sphere.

*Proof.* We first note that, since $\delta_{ji} = -\delta_{ij}$, $W_{ij}^{(s)} = W_{ji}^{(s)}$ for $i \neq j$, and $W_{ii}^{(s)} = 0$, we may express $\eta(\boldsymbol{\theta})$ (equation (2.1)) in terms of a sum over ordered pairs:

$$\tfrac{1}{2}\eta(\boldsymbol{\theta}) = \sum_{i<j} W_{ij}^{(s)} |e^{i\theta_i} - e^{i\delta_{ij}} e^{i\theta_j}|^2. \tag{3.5}$$

Then, distinguishing between the three different ways in which each $i$ and $j$ may be connected, we have

$$\tfrac{1}{2}\eta(\boldsymbol{\theta}) = \sum_{i<j:A_{ij}=1,A_{ji}=1} |e^{i\theta_i} - e^{i\theta_j}|^2 + \sum_{i<j:A_{ij}=1,A_{ji}=0} \tfrac{1}{2}|e^{i\theta_i} - e^{-i2\pi g} e^{i\theta_j}|^2 \tag{3.6}$$

$$+ \sum_{i<j:A_{ij}=0,A_{ji}=1} \tfrac{1}{2}|e^{i\theta_i} - e^{i2\pi g} e^{i\theta_j}|^2. \tag{3.7}$$

The likelihood $L$ of the graph $G$ from a model of the form (3.1)–(3.4) is given by

$$L(G) = \prod_{i<j:A_{ij}=1,A_{ji}=1} f(\theta_i, \theta_j) \prod_{i<j:A_{ij}=1,A_{ji}=0} q(\theta_i, \theta_j) \prod_{i<j:A_{ij}=0,A_{ji}=1} l(\theta_i, \theta_j)$$

$$\times \prod_{i<j:A_{ij}=0,A_{ji}=0} (1 - f(\theta_i, \theta_j) - q(\theta_i, \theta_j) - l(\theta_i, \theta_j)),$$

which we may rewrite as

$$L(G) = \prod_{i<j:A_{ij}=1,A_{ji}=1} \frac{f(\theta_i, \theta_j)}{1 - f(\theta_i, \theta_j) - q(\theta_i, \theta_j) - l(\theta_i, \theta_j)}$$

$$\times \prod_{i<j:A_{ij}=1,A_{ji}=0} \frac{q(\theta_i, \theta_j)}{1 - f(\theta_i, \theta_j) - q(\theta_i, \theta_j) - l(\theta_i, \theta_j)}$$

$$\times \prod_{i<j:A_{ij}=0,A_{ji}=1} \frac{l(\theta_j, \theta_i)}{1 - f(\theta_i, \theta_j) - q(\theta_i, \theta_j) - l(\theta_i, \theta_j)}$$

$$\times \prod_{i<j} (1 - f(\theta_i, \theta_j) - q(\theta_i, \theta_j) - l(\theta_i, \theta_j)).$$

The final factor on the right-hand side, which is the probability of the null graph, takes the same value for any $\boldsymbol{\theta} \in \mathbb{R}^n$ such that $\theta_i = v_{p_i}$, since each ordered pair of arguments appears exactly once. We may therefore ignore this factor when maximizing the likelihood. Then, taking the logarithm and negating, we see that maximizing the likelihood is equivalent to minimizing the expression

$$\sum_{i<j:A_{ij}=1,A_{ji}=1} \ln\left[\frac{1 - f(\theta_i, \theta_j) - q(\theta_i, \theta_j) - l(\theta_i, \theta_j)}{f(\theta_i, \theta_j)}\right] \tag{3.8}$$

$$+ \sum_{i<j:A_{ij}=1,A_{ji}=0} \ln\left[\frac{1 - f(\theta_i, \theta_j) - q(\theta_i, \theta_j) - l(\theta_i, \theta_j)}{q(\theta_i, \theta_j)}\right] \tag{3.9}$$

$$+ \sum_{i<j:A_{ij}=0,A_{ji}=1} \ln\left[\frac{1 - f(\theta_i, \theta_j) - q(\theta_i, \theta_j) - l(\theta_i, \theta_j)}{l(\theta_i, \theta_j)}\right]. \tag{3.10}$$

Comparing terms in (3.8)–(3.10) and (3.6)–(3.7), we see that the two minimization problems are equivalent if

$$\ln\left[\frac{1 - f(\theta_i, \theta_j) - q(\theta_i, \theta_j) - l(\theta_i, \theta_j)}{f(\theta_i, \theta_j)}\right] = \gamma|e^{i\theta_i} - e^{i\theta_j}|^2$$

$$= \gamma(2 - 2\cos(\theta_i - \theta_j)),$$

$$\ln\left[\frac{1 - f(\theta_i, \theta_j) - q(\theta_i, \theta_j) - l(\theta_i, \theta_j)}{q(\theta_i, \theta_j)}\right] = \frac{\gamma}{2}|e^{i\theta_i} - e^{-i2\pi g}e^{i\theta_j}|^2$$

$$= \gamma(1 - \cos(\theta_i - \theta_j + 2\pi g)),$$

and

$$\ln\left[\frac{1 - f(\theta_i, \theta_j) - q(\theta_i, \theta_j) - l(\theta_i, \theta_j)}{l(\theta_i, \theta_j)}\right] = \frac{\gamma}{2}|e^{i\theta_i} - e^{i2\pi g}e^{i\theta_j}|^2$$

$$= \gamma(1 - \cos(\theta_i - \theta_j - 2\pi g)),$$

where we may choose any positive constant $\gamma$ since the minimization problems are scale invariant. Solving for $f$, $q$ and $l$ as functions of $\theta_i$ and $\theta_j$ we arrive at the model in the statement of the theorem. ∎

For the model in theorem 3.1, the probability of an edge from node $i$ to node $j$ depends on the phase difference $\beta_{ij} = \theta_i - \theta_j$, the decay rate $\gamma$, and the parameter $g$. We see that $\gamma$ determines how rapidly the edge probability varies with the phase difference. In the extreme case when $\gamma = 0$, we obtain $f(\theta_i, \theta_j) = q(\theta_i, \theta_j) = l(\theta_i, \theta_j) = 1/4$, and thus the model reduces to a conditional Erdős–Rényi form. In addition, as $\gamma$ increases the graph generally becomes more sparse. This is because the likelihood of disconnection, $\exp[2\gamma(1 - \cos(\theta_i - \theta_j))]/Z_{ij}$, is greater than or equal to that of the other cases.

We note that having applied the magnetic Laplacian algorithm to estimate $\boldsymbol{\theta}$, there are two straightforward approaches to estimating $\gamma$. One way is to maximize the graph likelihood over $\gamma > 0$. Another is to choose $\gamma$ so that the expected edge density from the random graph model matches the edge density of the given network. We illustrate these approaches in §4.

**Remark 3.2.** Since the edge probabilities are functions of the phase differences and have a periodicity of $2\pi$, this model resembles the *periodic range-dependent random graph* (pRDRG) model in [14], which generates an undirected edge between $i$ and $j$ with probability $f(\min\{|j - i|, n - |j - i|\})$ for a given decay function $f$. We will therefore use the term *directed periodic range-dependent random graph* model (directed pRDRG) to describe the model in theorem 3.1.

## 3.2. The trophic range-dependent model

Now, given a set of trophic levels $\{h_i\}_{i=1}^n$, we define an unweighted, directed random graph model where

$$\mathbf{P}(A_{ij} = 1) = f(h_i, h_j) \tag{3.11}$$

and

$$\mathbf{P}(A_{ij} = 0) = 1 - f(h_i, h_j), \tag{3.12}$$

for some function $f$. Here, the probability of an edge $i \to j$ is independent of the probability of the edge $j \to i$.

Following our treatment of the directed pRDRG case, we are now interested in the inverse problem where we are given a graph and the model (3.11)–(3.12), and we wish to infer the trophic levels. We will assume that the trophic levels are to be assigned values from a discrete set $\{v_i\}_{i=1}^n$; that is, we must set $h_i = v_{p_i}$, where $p$ is a permutation vector. This setting includes the cases of assignment of nodes to trophic levels of specified size; for example, with $n = 12$, we could set $v_1 = v_2 = v_3 = 1$, $v_4 = v_5 = v_6 = 2$, $v_7 = v_8 = v_9 = 3$ and $v_{10} = v_{11} = v_{12} = 4$, in order to assign the nodes to four equal levels. Alternatively, $v_i = i$ would assign each node to its own level, which is equivalent to reordering the nodes. The following theorem shows that solving this type of inverse problem for suitable $f$ is equivalent to minimizing the trophic incoherence.

**Theorem 3.3.** *Suppose $h \in \mathbb{R}^n$ is constrained to take values such that $h_i = v_{p_i}$, where $p$ is a permutation vector. Then minimizing the trophic incoherence $F(\boldsymbol{h})$ in (2.3) over all such $\boldsymbol{h}$ is equivalent to maximizing the*

*likelihood that the graph came from a model of the form* (3.11)–(3.12) *in the case where*

$$f(h_i, h_j) = \frac{1}{1 + e^{\gamma(h_j - h_i - 1)^2}}$$

*for any positive $\gamma$.*

*Proof.* Noting that the denominator in (2.3) is independent of the choice of $\boldsymbol{h}$, this result is a special case of theorem 3.5 below, with $I(h_i, h_j) = (h_j - h_i - 1)^2$. ∎

For the model in theorem 3.3, the probability of an edge $i \to j$ is a function of the shifted, directed, squared difference in levels, $(h_j - h_i - 1)^2$. The larger this value, the lower the probability. Within the same level, where $h_i = h_j$, the probability is $1/(1 + e^{\gamma})$. The edge probability takes its maximum value of $1/2$ when $h_j - h_i = 1$, that is, when the edge starts at one level and finishes at the next highest level. We also see that the overall expected edge density is always smaller than $1/2$. Across different levels, where $h_i \neq h_j$, the edge $i \to j$ and the edge $j \to i$ are not generated with the same probability. If $|h_j - h_i - 1| < |h_i - h_j - 1|$, the edge $i \to j$ is more likely than $j \to i$. The two edge probabilities are equal if and only if $h_i = h_j$. Therefore, this model could be interpreted as a combination of an Erdős–Rényi model within the same level and a periodic range-dependent model across different levels.

The parameter $\gamma$ controls the decay rate of the likelihood as the shifted, directed, squared difference in levels increases. When $h_j - h_i = 1$, $\gamma$ plays no role. If $\gamma = 0$, the model reduces to Erdős–Rényi with an edge probability of $1/2$. As $\gamma \to \infty$, the edge probability tends to zero if $h_j - h_i \neq 1$. In this case, the model will generate a multipartite graph where edges are only possible in one direction between adjacent levels, and this happens with probability $1/2$. As mentioned previously in §3.1 and illustrated in §4, $\gamma$ can be fitted from a maximum likelihood estimate or by matching the edge density.

We note that the definition of trophic incoherence in (2.3) and the resulting trophic Laplacian algorithm make sense for a non-negatively weighted graph, in which case we have the following result. Here, to be concrete we assume that weights lie strictly between zero and one. Similar results can be obtained for weights from a discrete distribution.

**Theorem 3.4.** *Suppose $\boldsymbol{h} \in \mathbb{R}^n$ is constrained to take values such that $h_i = v_{p_i}$, where $p$ is a permutation vector. Then minimizing the trophic incoherence $F(\boldsymbol{h})$ in (2.3) over all such $\boldsymbol{h}$ for a weighted graph with weights in $(0, 1)$ is equivalent to maximizing the likelihood that the graph came from a model where each edge weight $A_{ij}$ is independent with density function*

$$f_{ij}(x) := \frac{1}{Z_{ij}\, e^{\gamma x (h_j - h_i - 1)^2}} \text{ for } x \in (0, 1), \quad \text{and } f(x) = 0 \text{ otherwise,} \tag{3.13}$$

*for any positive $\gamma$, where $Z_{ij} = (1 - e^{-\gamma(h_j - h_i - 1)^2})/(\gamma(h_j - h_i - 1)^2)$ is a normalization factor.*

*Proof.* This is a special case of theorem 3.6 below, where $I(h_i, h_j) = (h_j - h_i - 1)^2$. ∎

## 3.3. Generalized random graph model

The results in §§3.1 and 3.2 exploit the form of the objective function: the sum over all edges of a kernel function can be viewed as the sum of log-likelihoods. This shows that the minimization problem is equivalent to maximizing the likelihood of an associated random graph model, in the setting where we assign nodes to a discrete set of scalar values. The restriction to discrete values is used in the proofs to make the probability of the null graph constant. However, we emphasize that in practice the relaxed versions of the optimization problems, which are solved by the two algorithms, do not have this restriction. The magnetic Laplacian algorithm produces real-valued phase angles and the trophic Laplacian algorithm produces real-valued trophic levels.

We may extend the connection in theorem 3.3 to the case of higher dimensional node attributes, that is, where we wish to associate each node with a discrete vector from a set $\{v^{[k]}\}_{k=1}^n$, where each $v^{[k]} \in \mathbb{R}^d$ for some $d \geq 1$. This setting arises, for example, if we wish to visualize the network in higher dimension; a natural extension of the ring structure would be to place nodes at regularly spaced points on the surface of the unit sphere, see figure 2b, which we produced with the algorithm in [28]. The next result generalizes theorem 3.3 to this case.

**Theorem 3.5.** *Suppose we have an unweighted directed graph with adjacency matrix $A$ and a kernel function $I : \mathbb{R}^d \times \mathbb{R}^d \to \mathbb{R}_+$, and suppose that we are free to assign elements $\{h^{[k]}\}_{k=1}^n$ to values from the set $\{v^{[k]}\}_{k=1}^n$; that is,*

we allow $h^{[k]} = v^{[p_k]}$ where $p$ is a permutation vector. Then minimizing

$$\sum_{i,j} A_{ij} I(h^{[i]}, h^{[j]}) \tag{3.14}$$

over all such $\{h^{[k]}\}_{k=1}^n$ is equivalent to maximizing the likelihood that the graph came from a model where the (independent) probability of the edge $i \to j$ is

$$f(h^{[i]}, h^{[j]}) = \frac{1}{1 + e^{\gamma I(h^{[i]}, h^{[j]})}}, \tag{3.15}$$

for any positive $\gamma$.

*Proof.* Given $\{h^{[k]}\}_{k=1}^n$, the probability of generating a graph $G$ from the model stated in the theorem is

$$L(G) = \prod_{i,j : A_{ij}=1} f(h^{[i]}, h^{[j]}) \prod_{i,j : A_{ij}=0} \left(1 - f(h^{[i]}, h^{[j]})\right)$$

$$= \prod_{i,j : A_{ij}=1} \frac{f(h^{[i]}, h^{[j]})}{1 - f(h^{[i]}, h^{[j]})} \prod_{i,j} \left(1 - f(h^{[i]}, h^{[j]})\right).$$

The second factor on the right-hand side, the probability of the null graph, does not depend on the choice of $\{h^{[k]}\}_{k=1}^n$. So we may ignore this factor, and after taking logs and negating we arrive at the equivalent problem of minimizing

$$\sum_{i,j : A_{ij}=1} \ln\left[\frac{1 - f(h^{[i]}, h^{[j]})}{f(h^{[i]}, h^{[j]})}\right]. \tag{3.16}$$

Comparing (3.16) and (3.14), we see that two minimization problems have the same solution when

$$\ln\left[\frac{1 - f(h^{[i]}, h^{[j]})}{f(h^{[i]}, h^{[j]})}\right] = \gamma I(h^{[i]}, h^{[j]}),$$

for any positive $\gamma$, and the result follows. ∎

For the model in theorem 3.5, given $\{h^{[k]}\}_{k=1}^n$ the edge $i \to j$ appears according to a Bernoulli distribution with probability $f(h^{[i]}, h^{[j]})$, and hence with variance

$$f(h^{[i]}, h^{[j]})[1 - f(h^{[i]}, h^{[j]})] = \frac{e^{\gamma I(h^{[i]}, h^{[j]})}}{[1 + e^{\gamma I(h^{[i]}, h^{[j]})}]^2}.$$

When $I(h^{[i]}, h^{[j]}) = 0$ the probability is $1/2$ and the variance takes its largest value, $1/4$. The edge probability is symmetric about $i$ and $j$ if and only if the function $I$ is symmetric about its arguments. In the case of squared Euclidean distance, $I(h^{[i]}, h^{[j]}) = \|h^{[i]} - h^{[j]}\|^2$, and an undirected graph, the relaxed version of the minimization problem is solved by taking $d$ eigenvectors corresponding to the smallest eigenvalues of the standard graph Laplacian.

For completeness, we now state and prove a weighted analogue of theorem 3.5 assuming that weights lie strictly between zero and one. Discrete-valued weights may be dealt with similarly.

**Theorem 3.6.** *Suppose $\{h^{[k]}\}_{k=1}^n$ may take values from the given set $\{v^{[k]}\}_{k=1}^n$; that is, $h^{[k]} = v^{[p_k]} \in \mathbb{R}^d$, where $p$ is a permutation vector. Then, given a weighted graph with weights in $(0, 1)$, minimizing the expression (3.14) over all such $\{h^{[k]}\}_{k=1}^n$ is equivalent to maximizing the likelihood that the graph came from a model where $A_{ij}$ has (independent) density*

$$f_{ij}(x) = \frac{1}{Z_{ij}} e^{\gamma x I(h^{[i]}, h^{[j]})}, \quad \text{for } x \in (0, 1), \quad \text{and } f(x) = 0 \text{ otherwise}, \tag{3.17}$$

*for any positive $\gamma$, where*

$$Z_{ij} = \frac{1 - e^{-\gamma I(h^{[i]}, h^{[j]})}}{\gamma I(h^{[i]}, h^{[j]})}$$

*is a normalization factor.*

*Proof.* It is straightforward to check that the normalization factor $Z_{ij}$ ensures

$$\int_{y=0}^{1} f_{ij}(y)\,\mathrm{d}y = 1.$$

Now the product over all pairs $\prod_{i,j} Z_{ij}$ is independent of the choice of permutation vector $p$. Hence, under the model defined in the theorem, maximizing the likelihood of the graph $G$ is equivalent to maximizing $\prod_{i,j} f_{ij}(A_{ij})$. After taking logarithms and negating, we see that the choice (3.17) allows us to match (3.14). ∎

**Remark 3.7.** It is natural to ask whether the frustration (2.1) fits into the form (3.14), and hence has an associated random graph model of the form (3.15). We see from (3.5) that the frustration may be written

$$\eta(\boldsymbol{\theta}) = \sum_{i,j} A_{ij}|e^{\mathbf{i}\theta_i} - e^{\mathbf{i}\delta_{ij}}e^{\mathbf{i}\theta_j}|^2.$$

However, the factor $|e^{\mathbf{i}\theta_i} - e^{\mathbf{i}\delta_{ij}}e^{\mathbf{i}\theta_j}|^2$ depends (through $\delta_{ij}$) on $A_{ij}$, and hence we do not have an expression of the form (3.14). This explains why a new type of model, with conditional dependence between the $i \to j$ and $j \to i$ connections, was needed for theorem 3.1.

## 3.4. Model comparison

The random graph models appearing in §3 capture the characteristics of linear and periodic directed hierarchies. Hence it may be of interest (a) to analyse properties of these models and (b) to use these models to evaluate the performance of computational algorithms. However, in the remainder of this work we focus on a follow-on topic of more direct practical significance. The magnetic Laplacian and trophic Laplacian algorithms allow us to compute node attributes $\boldsymbol{\theta}$ and $h$ in $\mathbb{R}^n$ for a given graph, leading to unsupervised node ordering. The main computation required in this step is finding dominant eigenvector–eigenvalue pairs. Assuming that the network is sparse (each node has an $O(1)$ degree) and that the power method gives the required accuracy in a finite number of iterations, this is an $\mathcal{O}(n)$ computation. Motivated by theorems 3.1 and 3.3, we may then compute the likelihood of the graph for this choice of attributes, which has a complexity of $\mathcal{O}(n^2)$. By comparing likelihoods, we may quantify which underlying structure is best supported by the data. An extra consideration is that both random graph models involve a free parameter, $\gamma > 0$, which is needed to evaluate the likelihood. As discussed earlier, one option is to fit $\gamma$ to the data, for example by matching the expected edge density from the model with the edge density of the given graph. However, based on our computational tests, we found that a more reliable approach was to choose the $\gamma$ that maximizes the likelihood, once the node attributes were available; see §§4 and 5 for examples. Our overall proposed workflow for model comparison is summarized in algorithm 3.

---

**Algorithm 3.** Model comparison.

---

**Result:** Comparison of possible graph structures

**Input adjacency matrix** $A$;

**for** *Candidate spectral methods* **do**

 **Compute node attributes** (in our case with magnetic and trophic Laplacian algorithms);

 **Derive the associated random graph model**;

 **Calculate maximum likelihood over** $\gamma > 0$;

**end**

**Report or compare maximum likelihoods**

---

# 4. Results on synthetic networks

In this section, we demonstrate the model comparison workflow on synthetic networks. These networks are generated using the directed pRDRG model and the trophic RDRG model. Hence, we have a 'ground truth' concerning whether a network is more linear or periodic. Note that the magnetic Laplacian algorithm and associated random graph model have a parameter $g$ that controls the spacing between

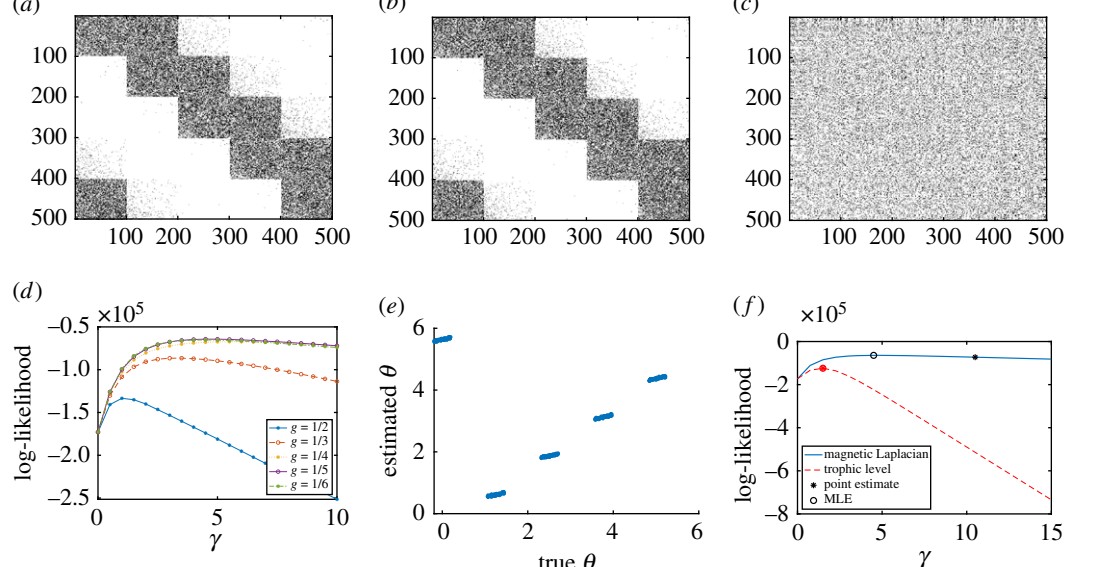

**Figure 3.** Magnetic Laplacian and trophic Laplacian algorithms applied to a synthetic directed pRDRG. (*a*) Input adjacency matrix, (*b*) magnetic Laplacian reordering, (*c*) trophic Laplacian reordering, (*d*) likelihood of directed pRDRG, (*e*) estimated $\theta$ and (*f*) model comparison.

clusters. Therefore, when using the magnetic Laplacian algorithm our first step is to select the parameter $g$ based on the maximum likelihood of the graph.

## 4.1. Directed pRDRG model

We generate a synthetic network using the directed pRDRG model with $K$ clusters of size $m$, and hence $n = m\,K$ nodes. An array of angles $\boldsymbol{\theta} \in \mathbb{R}^n$ is created, forming evenly spaced clusters $C_1, C_2, \ldots, C_K$. This is achieved by letting $\theta_i = (2\pi(l-1)/K) + \sigma$ if $i \in C_l$, where $\sigma \sim \text{unif}(-a, a)$ is added noise. We then construct the adjacency matrix according to the probabilities in theorem 3.1 with $g = 1/K$. We choose $m = 100$, $K = 5$, $\gamma = 5$ and $a = 0.2$ and the corresponding adjacency matrix is shown in figure 3*a*.

The magnetic Laplacian algorithm is then applied to the adjacency matrix to estimate phase angles and reorder the nodes. The reordered adjacency matrix (figure 3*b*) recovers the original structure. The trophic Laplacian algorithm is also applied to estimate the trophic level of each node. Figure 3*c* shows the adjacency matrix reordered by the estimated trophic levels, which hides the original pattern. Intuitively, the trophic Laplacian algorithm is unable to distinguish between these nodes since there is no clear 'lowest' or 'highest' level among the directed clusters.

Figure 3*d* illustrates how the optimal parameter $g$ is selected. The plots show the likelihood that the network is generated by a directed pRDRG model for $g = \frac{1}{2}, \frac{1}{3}, \frac{1}{4}, \frac{1}{5}, \frac{1}{6}$ assuming we are interested in structures with at most 6 directed clusters. We see that $g = \frac{1}{5}$ has the highest maximum likelihood, as expected. Consequently, we choose $g = 1/5$ for the magnetic Laplacian algorithm. In addition for this value of $g$, we plot in figure 3*e* the phase angles estimated with the magnetic Laplacian algorithm against the true phase angles. The linear relationship confirms that the algorithm recovers the five clusters in the presence of noise.

We finally in figure 3*f* compare the likelihood of a directed pRDRG against the likelihood of a trophic RDRG. Both likelihoods are calculated using several test points for $\gamma$. The highest points are highlighted with circles and they correspond to the maximum-likelihood estimators (MLE) for $\gamma$. Not surprisingly, in this case, the magnetic Laplacian algorithm achieves a higher maximum. Asterisks highlight the point estimates arising when the expected number of edges is matched to the actual number of edges. We see here, and also observed in similar experiments, that the maximum-likelihood estimate for $\gamma$ produces a more accurate result. We also found (numerical experiments not presented here) that the accuracy of both types of $\gamma$ estimates improves as $n$ increases when using the magnetic Laplacian algorithm.

## 4.2. The trophic RDRG model

Following on from the previous subsection, we now generate synthetic data by simulating the trophic RDRG model with levels $C_1, C_2, \ldots, C_K$, where each level has $m$ nodes. In particular, we generate an array of trophic indices $\boldsymbol{h} \in \mathbb{R}^n$, where the total number of nodes is $n = m\,K$. We let $h_i = l + \sigma$ if $i \in C_l$

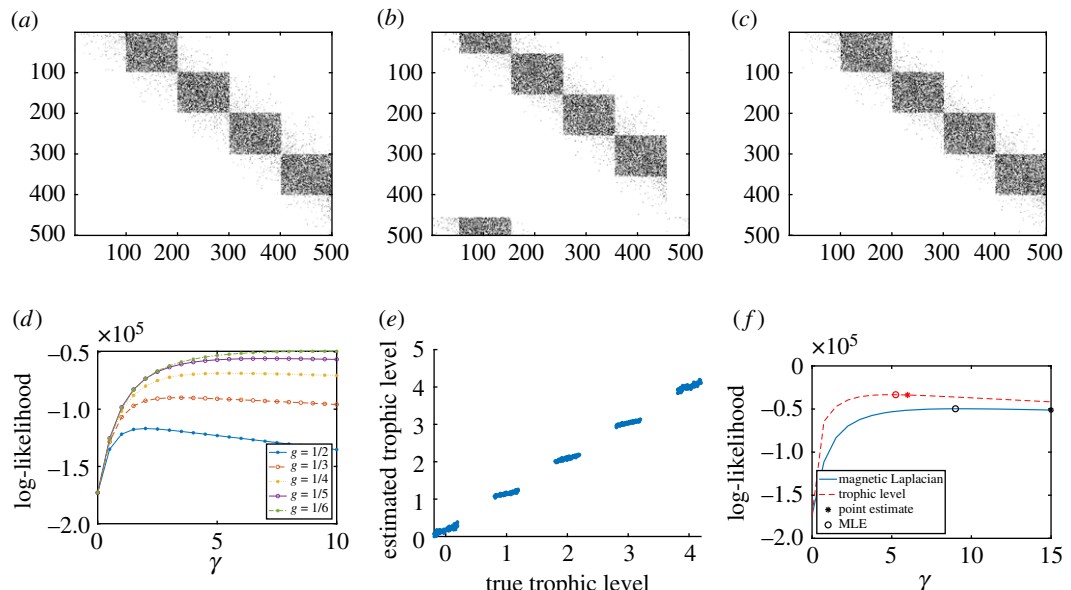

**Figure 4.** Magnetic Laplacian and trophic Laplacian algorithms applied to a synthetic trophic RDRG. (*a*) Input adjacency matrix, (*b*) magnetic Laplacian reordering, (*c*) trophic Laplacian reordering, (*d*) likelihood of directed pRDRG, (*e*) estimated trophic level, (*f*) model comparison.

for $1 \leq l \leq K$, where $\sigma \sim \mathrm{unif}(-a, a)$ is added noise. The edges are then generated according to the probabilities in theorem 3.3. In the following example, we use $K = 5$, $m = 100$, $a = 0.2$ and $\gamma = 5$. This generates a network with five clusters forming a linear directed flow, as shown in figure 4*a*.

We see in figure 4*c* that the trophic Laplacian algorithm recovers the underlying pattern. Figure 4*b* shows that the magnetic Laplacian algorithm also gives adjacent locations to nodes in the same cluster, and places the clusters in order, modulo a 'wrap-around' effect that arises due to its periodic nature. Figure 4*d* suggests that the optimal magnetic Laplacian parameter is $g = 1/6$. For this case, it is reasonable that $g = 1/K$ is not identified, since the disconnection between the first and the last cluster contradicts the structure of the directed pRDRG model.

The trophic levels estimated using the trophic Laplacian are consistent with the true trophic levels, as shown by the linear pattern in figure 4*e*. As expected, the trophic Laplacian produces a higher maximum likelihood for this network (figure 4*f*) and a more accurate MLE and point estimate for $\gamma$. We observe (in similar experiments not presented here) that when using the trophic Laplacian, the accuracy of both estimates increases using the trophic Laplacian.

# 5. Results on real networks

We now discuss practical use cases for the model comparison tool on a range of real networks. We emphasize that the tool is not designed to discover whether a given directed network has linear or directed hierarchical structure; rather it aims to quantify which of the two structures is best supported by the data in a relative sense. Since both models under investigation assume no self-loops, we discard these if they are present in the data. Following common practice, we also preprocess by retaining the largest strongly connected component to emphasize directed cycles. This ensures that any pair of nodes can be connected through a sequence of directed edges. However, when the strongly connected component contains too few nodes, we analyse the largest weakly connected component instead.

We give details on four networks, covering examples of the two cases where linear and periodic structure dominates. For the first two networks, we show network visualizations to illustrate the results further. In §5.5, we present summary results over 15 networks.

## 5.1. Food web

In the Florida Bay food web[1] [29], nodes are components of the system, and unweighted directed edges represent carbon transfer from the source nodes to the target nodes [30], which usually means that the

[1]https://snap.stanford.edu/data/Florida-bay.html.

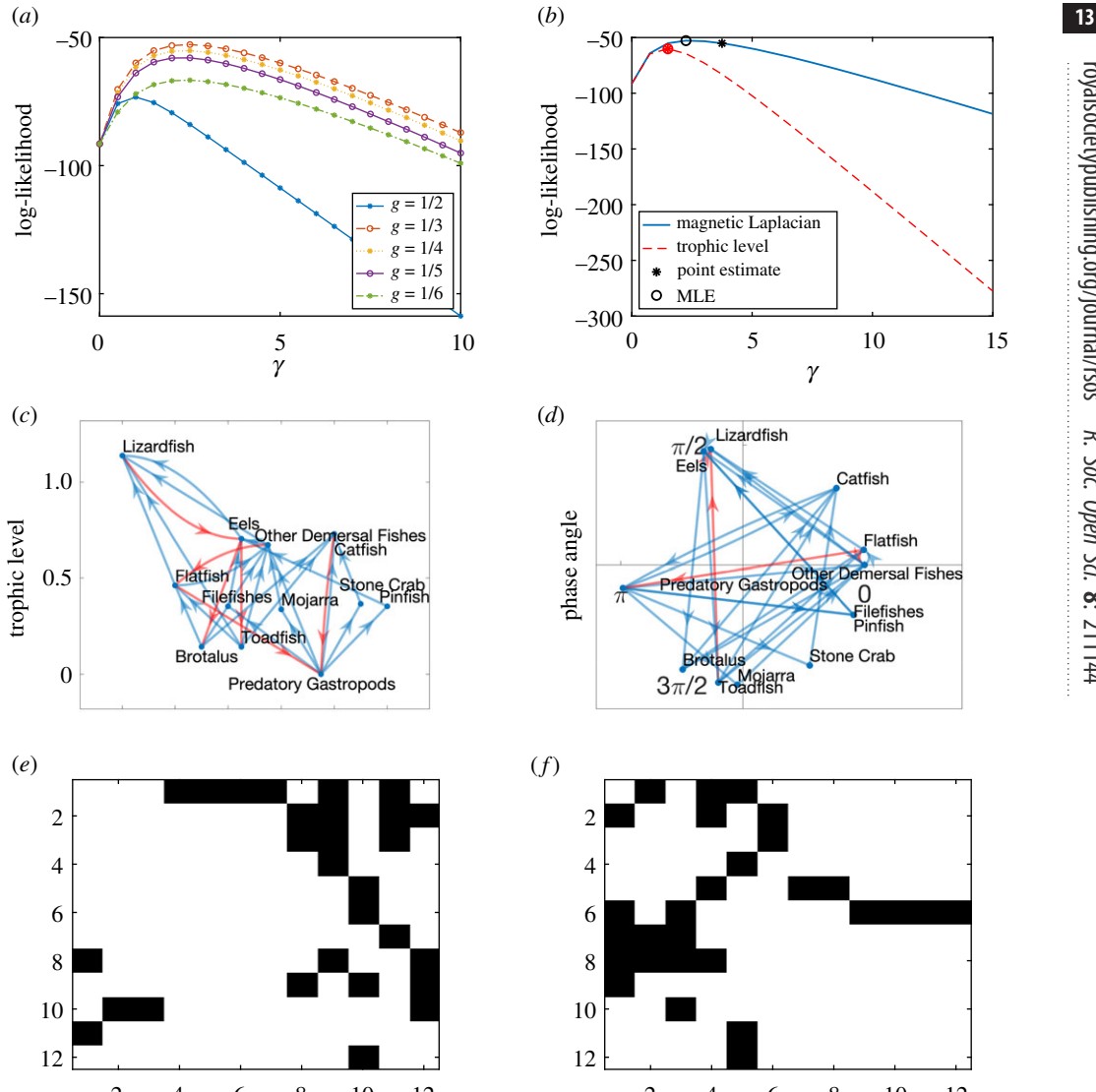

**Figure 5.** Results for the Florida Bay food web. (a) Likelihood of directed pRDRG, (b) model comparison, (c) estimated trophic level, (d) magnetic eigenmap, (e) trophic Laplacian reordering and (f) magnetic Laplacian reordering.

latter feed on the former. Besides organisms, the nodes also contain non-living components, such as carbon dissolved in the water column. Since we are more interested in the relationship between organisms, we remove those non-living components from the network. We analyse the largest strongly connected component of the network, which comprises 12 nodes and 28 edges.

We estimate the phase angles of each node using the magnetic Laplacian algorithm based on the optimal choice $g = 1/3$ (figure 5a). Figure 5b compares the likelihood of the food web being generated by the directed pRDRG model with the likelihood of it being generated by the trophic RDRG model, as $\gamma$ varies. The directed pRDRG model achieves a higher maximum likelihood, suggesting that the structure is more periodic than linear. In figure 5c, the heights of the nodes correspond to their estimated trophic levels on a vertical axis. We see that 22 edges point upwards, these are shown in blue. There are six downward edges, highlighted in red, which violate the trophic structure. The magnetic Laplacian mapping in figure 5d arranges 26 edges in a counterclockwise direction, shown in blue, with 2 edges, shown in red, violating the structure and pointing in the reverse orientation.

With $g = 1/3$, the magnetic Laplacian mapping is encouraging cycles in the food chain, and these are visible in figure 5d, notably between members of three categories: (i) flatfish and other demersal fishes; (ii) lizardfish and eels; and (iii) toadfish and brotalus. Another noticeable distinction is that the magnetic Laplacian mapping positions eels close to lizardfish, and flatfish near other demersal fishes by accounting for the reciprocal edges, while the trophic Laplacian mapping places them further apart. In figure 5e,f, we show the reordered adjacency matrix arising from the two algorithms.

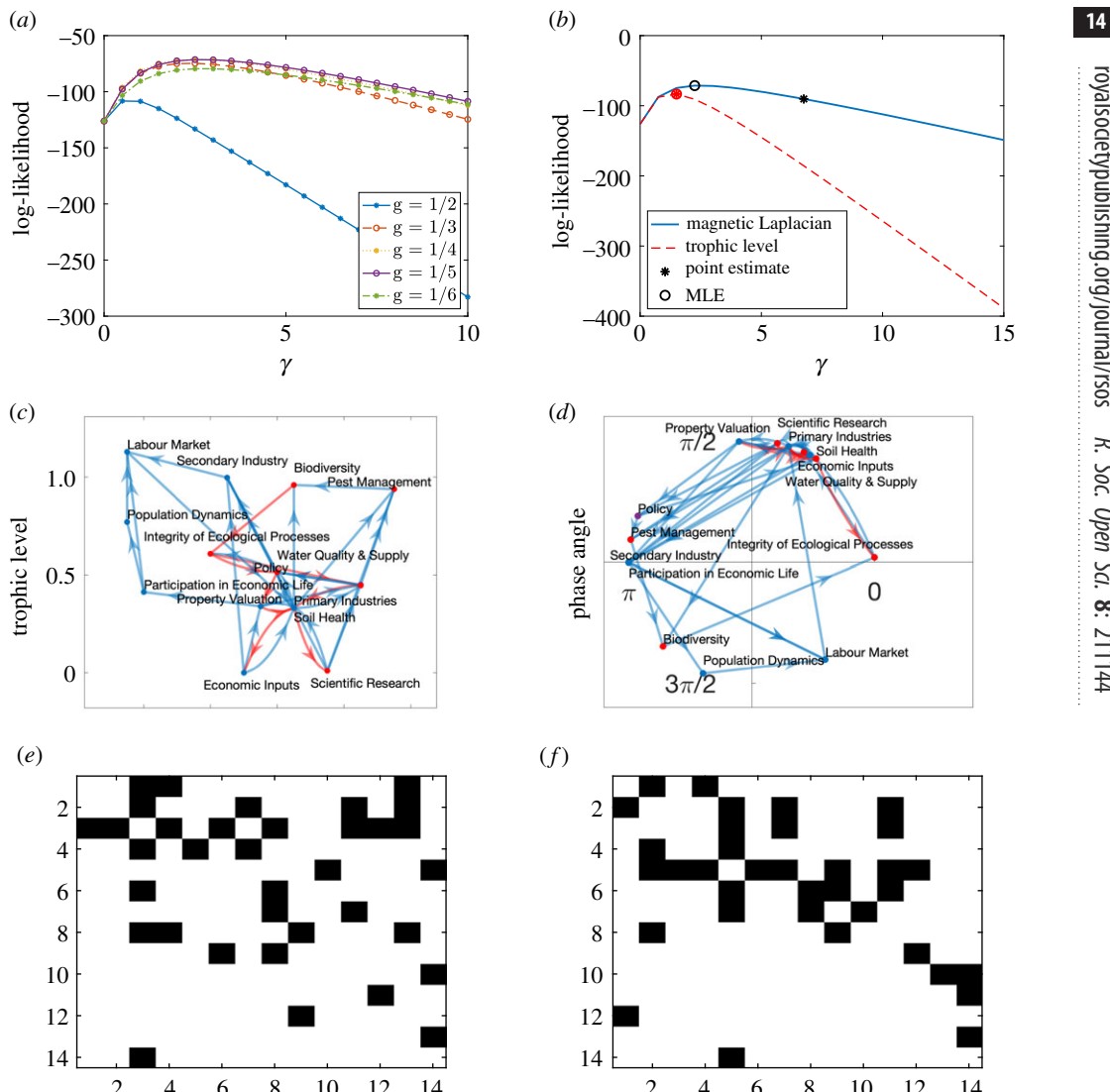

**Figure 6.** Results for the Motueka catchment influence matrix. (a) Likelihood of directed pRDRG, (b) model comparison, (c) estimated trophic level, (d) magnetic eigenmap, (f) trophic Laplacian reordering and (g) magnetic Laplacian reordering.

## 5.2. Influence matrix

The influence matrix we study quantifies the influence of selected system factors in the Motueka Catchment of New Zealand [31]. The original influence matrix consists of integer scores between 0 and 5, measuring to what extent the row factors influence the column factors, where a bigger value represents a stronger impact. The system factors and influence scores were developed by pooling the views of local residents. To convert to an unweighted network, we binarize the weights by keeping only the edges between each factor and the factor(s) it influences most strongly. We then select the largest strongly connected component, which comprises 14 nodes and 35 edges.

The optimal parameter for the magnetic Laplacian is $g = 1/4$ (figure 6a). The mapping from the magnetic Laplacian has a higher maximum likelihood than the trophic Laplacian mapping, indicating a more periodic structure (figure 6b). The trophic Laplacian mapping in figure 6c aims to reveal a hierarchical influence structure. Here, scientific research and economic inputs are assigned lower trophic levels, suggesting that they are the fundamental influencers. The labour market is placed at the top, indicating that it tends to be influenced by other factors. However, there are eight edges, highlighted in red, that point downwards, violating the directed linear structure.

On the other hand, the magnetic Laplacian mapping in figure 6d aims to reveal four directed clusters with phase angles of approximately 0, $\pi/2$, $\pi$, $3\pi/2$. We highlight the nodes corresponding to ecological factors in red and socio-economic factors in blue. The cluster near $\pi/2$ with 6 nodes contains a combination of ecological and socio-economic factors, and includes 6 reciprocal edges between

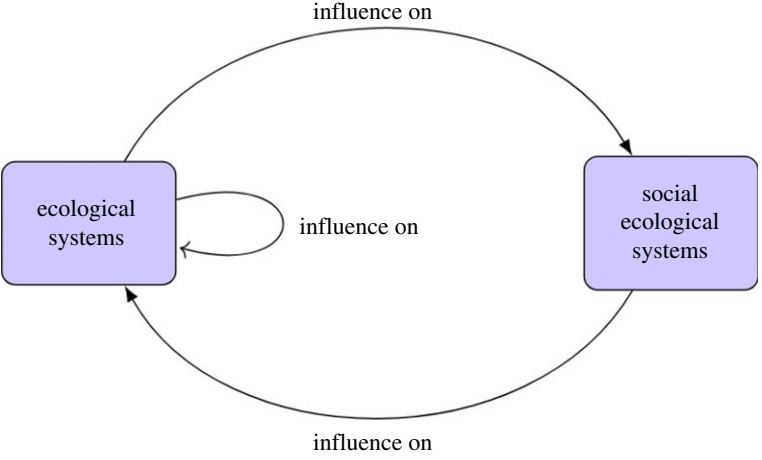

**Figure 7.** Influence matrix schematic graph, based on [31, fig. 5*a*].

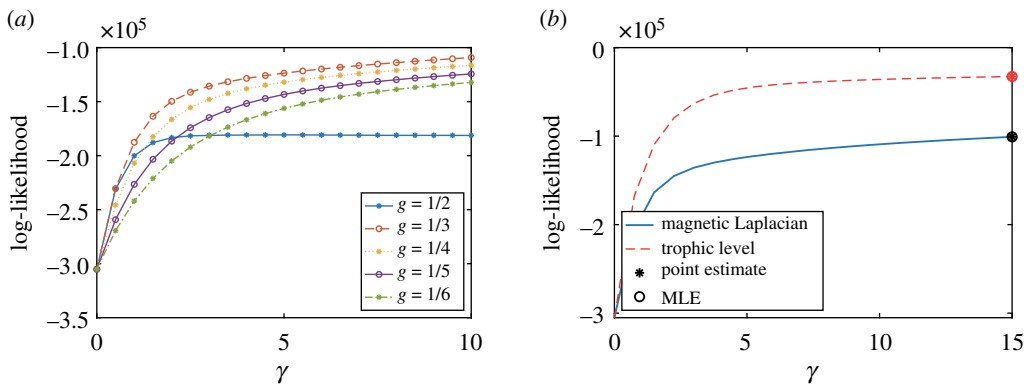

**Figure 8.** Results for a yeast transcriptional regulation network. (*a*) Likelihood of directed pRDRG and (*b*) model comparison.

ecological factors and socio-economic factors. Adjacency matrix reorderings are shown in figure 6*f,g*. Overall, the pattern agrees with the conceptual schematic model proposed in [31, fig. 5*a*], which we have reproduced in figure 7. This model posits that ecological factors exert influence on socio-economic factors, which in turn influence ecological factors, while the ecological system also influences itself.

## 5.3. Yeast transcriptional regulation network

We now analyse a gene transcriptional regulation network[2] [29] for a type of yeast called *S. cerevisiae* [32], where a node represents an operon made up of a group of genes in mRNA. An edge from operon $i$ to $j$ indicates that the transcriptional factor encoded by $j$ regulates $i$. The original network is directed and signed, with signs indicating activation and deactivation. Here, we ignore the signs and only consider the connectivity pattern. Since the largest strongly connected component has very few nodes, we take the largest weakly connected component, which comprises 664 nodes and 1078 edges.

This is a very sparse network and consequently the log-likelihood of the directed pRDRG (figure 8*a*) keeps increasing as a function of the decay rate parameter $\gamma$ in the range we tested. We select $g = 1/3$ as the optimal parameter for the magnetic Laplacian, and compare the log-likelihood of two models in figure 8*b*. This time the trophic version achieves a higher maximum likelihood, favouring a linear structure.

## 5.4. *Caenorhabditis elegans* frontal neural network

*Caenorhabditis elegans* is the only organism whose neural network has been fully mapped. The neural network of *C. elegans*[3] [29] is unweighted and directed, representing connections between neurons and

[2]http://snap.stanford.edu/data/S-cerevisiae.html.

[3]http://snap.stanford.edu/data/C-elegans-frontal.html.

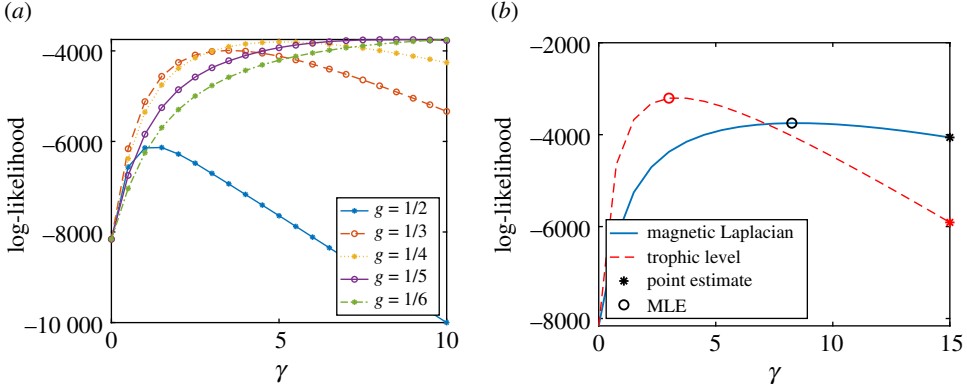

**Figure 9.** *Caenorhabditis elegans* frontal neural network. (*a*) Likelihood of directed pRDRG and (*b*) model comparison.

**Table 1.** Comparison summary statistics. Periodic (linear) directed structure is found to be preferred for networks in the first 8 (last 7) rows.

| dataset | nodes | edges | g | $\ln(P_{pRDRG}/P_{Trophic})$ |
|---|---|---|---|---|
| directed pRDRG (s) | 500 | 49277 | 1/5 | $5.99 \times 10^{+04}$ |
| food web (s) [30] | 12 | 28 | 1/3 | $1.17 \times 10^{+01}$ |
| influence matrix (s) [31] | 14 | 35 | 1/4 | $1.72 \times 10^{+01}$ |
| US migration (s)[a] | 51 | 729 | 1/6 | $5.03 \times 10^{+02}$ |
| US IO (s)[b] | 31 | 299 | 1/6 | $5.67 \times 10^{+01}$ |
| trade (s)[c] | 17 | 85 | 1/6 | $2.02 \times 10^{+01}$ |
| transportation (s)[d] [29,35] | 456 | 71959 | 1/6 | $4.66 \times 10^{+04}$ |
| flight (s)[e] | 227 | 23113 | 1/6 | $7.22 \times 10^{+03}$ |
| trophic level graph (w) | 500 | 19956 | 1/6 | $-1.63 \times 10^{+04}$ |
| *C. elegans* (s) [33] | 109 | 637 | 1/6 | $-4.74 \times 10^{+02}$ |
| yeast (w) [32] | 664 | 1078 | 1/3 | $-6.46 \times 10^{+04}$ |
| political blog (s)[f] [36] | 793 | 15781 | 1/5 | $-3.42 \times 10^{+04}$ |
| shopping basket (w)[g] | 27 | 84 | 1/6 | $-1.35 \times 10^{+02}$ |
| venue reopen (w) [37] | 13 | 19 | 1/6 | $-1.82 \times 10^{+01}$ |
| word adjacency (w)[f] [38] | 112 | 425 | 1/6 | $-8.21 \times 10^{+02}$ |

[a]https://www.census.gov/content/census/en/library/publications/2003/dec/censr-8.html.
[b]https://stats.oecd.org/Index.aspx?DataSetCode=IOTSI4_2018.
[c]http://www.economicswebinstitute.org/worldtrade.htm.
[d]http://snap.stanford.edu/data/reachability.html.
[e]https://www.visualizing.org/global-flights-network/.
[f]http://www-personal.umich.edu/mejn/netdata/.
[g]https://www.dunnhumby.com/source-files/.

synapses [33]. We investigate its largest strongly connected component with 109 nodes and 637 edges. The optimal value for the parameter $g$ among the test points is $g = 1/5$ (figure 9*a*). The trophic Laplacian algorithm achieves a higher maximum likelihood than the magnetic Laplacian algorithm using figure 9*b*. This preference for a linear directed structure is consistent with the tube-like shape of the organism [34].

## 5.5. Other real networks

A summary of further real-world network comparisons is given in table 1.

In the dataset column, we use (s) and (w) to indicate whether the largest *strongly* or *weakly* connected component is analysed, respectively. The fourth column specifies the optimal parameter $g$ for the magnetic Laplacian determined through grid search among the test points $g = 1/2, 1/3, 1/4, 1/5, 1/6$. The decay parameter $\gamma$ used for the grid search ranges from 0 to 20 with a step size of 0.5. The last column shows the logarithm of the ratio between the maximum likelihoods of the directed pRDRG and trophic models. Hence, periodic/linear structure is seen to be favoured for the networks in the first 8 rows/last 7 rows.

# 6. Discussion

Spectral methods can be used to extract structures from directed networks, allowing us to detect clusters, rank nodes and visualize patterns. This work exploited a natural connection between spectral methods for directed networks and generative random graph models. We showed that the magnetic Laplacian and tropic Laplacian can each be associated with a range-dependent random graph. In the magnetic Laplacian case, the new random graph model has the interesting property that the probabilities of $i \rightarrow j$ and $j \rightarrow i$ connections are not independent. Our theoretical analysis provided a workflow for quantifying the relative strength of periodic versus linear directed hierarchy, using a likelihood ratio, adding value to the standard approach of visualizing a new graph layout or reordering the adjacency matrix.

We demonstrated the model comparison workflow on synthetic networks, and also showed examples where real networks were categorized as more linear or periodic. The results illustrate the potential for the approach to reveal interesting patterns in networks from ecology, biology, social sciences and other related fields.

There are several promising directions for related future work. It would be of interest to use the likelihood ratios to compare this network feature across a well-defined category in order to address questions such as 'are results between top chess players more or less periodic than results between top tennis players?' and 'does an organism that is more advanced in an evolutionary sense have more periodic connectivity in the brain?' An extension of the comparison tool to weighted networks should also be possible; here there are notable, and perhaps application-specific, issues about how to generalize and interpret the magnetic Laplacian. Also, the comparison could be extended to include other types of structure, including stochastic block and core–periphery versions [39]. This introduces further challenges of (a) accounting for different numbers of model parameters and (b) dealing with nonlinear spectral methods. Furthermore, by introducing an appropriate null model it may be possible to quantify the presence of linear or periodic hierarchies in absolute, rather than relative, terms.

Data accessibility. This research made use of public domain data that are available from the Internet, as indicated in the text. Code for the experiments is available at https://github.com/OpalGX/Directed-Network-Laplacians.

Authors' contributions. X.G. carried out the numerical experiments and drafted the manuscript. All authors contributed to the theoretical research, the design of numerical experiments and the completion of the manuscript. All authors have read and approved the manuscript and gave final approval for publication.

Competing interests. The authors declare that there is no conflict of interest.

Funding. X.G. acknowledges support of MAC-MIGS CDT Scholarship under EPSRC grant no. EP/S023291/1. D.J.H. was supported by EPSRC Programme grant no. EP/P020720/1.

Acknowledgements. The authors thank Colin Singleton from the CountingLab for suggesting the Dunnhumby data used in table 1 and providing advice on data analysis.

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
