## [Peer Review File · Royal Society Open Science]

Review History

RSOS-211144.R0 (Original submission)

Review form: Reviewer 1

Is the manuscript scientifically sound in its present form?

Yes

Are the interpretations and conclusions justified by the results?

Yes

Is the language acceptable?

Yes

Do you have any ethical concerns with this paper?

No

Have you any concerns about statistical analyses in this paper?

No

Recommendation?

Accept with minor revision (please list in comments)

Comments to the Author(s)

The main focus of this work is on the development of Laplacian-based methods for uncovering clustered hierarchical structure in directed networks. Compared to the undirected counterpart, the directed case has received a lot less attention in the recent and distant literature, though a number of very recent works in this area have attracted significant attention, due to their applicability to various real world settings. From this perspective, the present piece of work is very timely, and a good addition to the literature.

The two algorithms considered are not new, but rather direct consequences of previous works in [5] and [7]. The authors could clarify better/earlier in the paper what exactly is meant by "establish connections between members of these random graph classes and algorithms from [7] and [5]" (as in the second bullet of the contribution). Should the reader understand that, in some sense, some reverse engineering is being done, by taking two established algorithms and finding suitable probabilistic models which they do a good job solving?

A set of theoretical results are provided to quantify the relative strength of the two possible structures: periodic versus linear directed hierarchy, using a likelihood ratio. The authors could discuss the case when neither structure is inherent in the data, and means by which one can draw such a conclusion. Some guidance could be provided for how one might design a statistical significance test for a given uncovered structure (so not for the pairwise comparison).

My only main concern is the lack of comparison with other existing methods applicable to the task at hand - the authors cite a number of relevant papers/algorithms from the clustering and ranking literatures, but do not compare against any of them.

A good number of real world data sets have been employed (though quite a few of the networks have an unusually small number of nodes). It would have been good to see performance on at least one large data set (with thousands of nodes, at least).

Spectral methods are known to underperform in the sparse regime; it would be good to assess the sensitivity of the two proposed methods to sparsity in the measurement graph, in particular the regime where the sampling probability is below $\log(n)/n$. Meaningful experiments in this context typically have the number of nodes at least 500-1000.

It would also be useful to place the chosen synthetic models in the context of stochastic block models; the input adj matrices in Fig 3 and 4 can also be seen as instances of stochastic block models, so it can be a bit confusing to see in the Conclusion that "the comparison could be extended to include other types of structure, including stochastic block versions"

Figure 1a would be more clear if it had on top the latent/meta-graph (a triangle with directed edges in a cycle, encoding the cluster structure, with each cluster being a meta-node in the meta-graph). And same for Figure 1b.

It would be good to see similar heatmaps as in Figure 3 & 4 (with ordered adj matrices) for some of the real data sets, as well.

Review form: Reviewer 2

Is the manuscript scientifically sound in its present form?

Yes

Are the interpretations and conclusions justified by the results?

Yes

Is the language acceptable?

Yes

Do you have any ethical concerns with this paper?

No

Have you any concerns about statistical analyses in this paper?

No

Recommendation?

Accept with minor revision (please list in comments)

Comments to the Author(s)

The manuscript introduces a random graph based model selection framework for spectral methods that can detect the relative importance of periodic and linear hierarchies in directed networks. It is clearly and nicely written, first providing theoretical grounding for the authors' contributions, then illustrating the methods and framework on synthetic and real data. I recommend the manuscript for publication subject to minor revisions as below:

- Certain parts in the motivation section 1 would benefit from further elaboration. When the authors state "because there is a greater variety of possible structures", it would be good to add a sentence to substantiate this. When they state "This hierarchy may be periodic or linear, depending on whether there are well-defined start and end groups", I think more detail is needed to explain the concepts "periodic" and "linear" intuitively to the reader and to motivate why/when they're important in real-world applications. These concepts are quite key to the paper. The authors could then perhaps link back to this in the real world section. A minor point in figure 1 is to explicitly state that colours represent clusters in the caption.

- The generality of the model selection contribution of the paper should be made clearer. In the abstract the authors state "We develop a general framework that allows us to associate methods based on optimization formulations with maximum likelihood problems on random graphs." This sounds potentially very useful, as most methods for detecting mesoscale structure are based on optimization formulations. However, the generality of the framework doesn't quite come through later on, e.g. when the authors list their main contributions in the motivation section. Is the framework only application to spectral methods? If so, why? In any case, it would be helpful to the reader if the authors were clearer about precisely how "general" the framework is both in the motivation and the conclusion.

- In bullet point 1 of contributions, perhaps "we propose two new classes" would be more appropriate so that "both" in bullet point three can be easily linked back to that. Again in bullet point 1, it would be good to include a citation after "unusual property". More generally, the main contributions bullet points could use some polish.

- 2b: can the authors give a stronger motivation for why they specifically focus on the Magnetic Laplacian algorithm and the Trophic Laplacian algorithm? The discussion of other related techniques is rather "in passing", and there are many techniques not mentioned at all. I am not suggesting anything too detailed, but it would be good if the authors offered some convincing motivation for their focus.

- In terms of practical use, it would be good to make it clear if the methods are supervised (e.g. number of clusters assumed to be known) or not, and how well they scale with the size of the network.
- In the conclusion the authors briefly mention a potential extension to weighted networks, can more be said here and/or can the authors connect what they state to the extensions presented in the manuscript (e.g. to weights in $(0,1)$)? Can anything be said about other types of weights (e.g. integers)?
- Minor points: on p. 13 the figures loaded with some difficulty (worth double checking) and in the authors' public Git repo it would be good to include further detail in the readme file. Eg the authors could perhaps clarify which function corresponds to which method/model/ model selection framework.

Decision letter (RSOS-211144.R0)

Dear Miss Gong,

On behalf of the Editors, we are pleased to inform you that your Manuscript RSOS-211144 "Directed Network Laplacians and Random Graph Models" has been accepted for publication in Royal Society Open Science subject to minor revision in accordance with the referees' reports. Please find the referees' comments along with any feedback from the Editors below my signature.

Please submit your revised manuscript and required files (see below) no later than 7 days from today's (ie 31-Aug-2021) date. Note: the ScholarOne system will 'lock' if submission of the revision is attempted 7 or more days after the deadline. If you do not think you will be able to meet this deadline please contact the editorial office immediately.

on behalf of Dr Robert MacKay (Associate Editor) and Mark Chaplain (Subject Editor)
 openscience@royalsociety.org

Associate Editor Comments to Author (Dr Robert MacKay):

Both reviewers are positive about the paper but ask for minor revisions. Their points are good, so I recommend publication subject to minor revision to take into account their comments.

Reviewer comments to Author:

Reviewer: 1

Comments to the Author(s)

The main focus of this work is on the development of Laplacian-based methods for uncovering clustered hierarchical structure in directed networks. Compared to the undirected counterpart, the directed case has received a lot less attention in the recent and distant literature, though a number of very recent works in this area have attracted significant attention, due to their applicability to various real world settings. From this perspective, the present piece of work is very timely, and a good addition to the literature.

The two algorithms considered are not new, but rather direct consequences of previous works in [5] and [7]. The authors could clarify better/earlier in the paper what exactly is meant by “establish connections between members of these random graph classes and algorithms from [7] and [5]” (as in the second bullet of the contribution). Should the reader understand that, in some sense, some reverse engineering is being done, by taking two established algorithms and finding suitable probabilistic models which they do a good job solving?

A set of theoretical results are provided to quantify the relative strength of the two possible structures: periodic versus linear directed hierarchy, using a likelihood ratio. The authors could discuss the case when neither structure is inherent in the data, and means by which one can draw such a conclusion. Some guidance could be provided for how one might design a statistical significance test for a given uncovered structure (so not for the pairwise comparison).

My only main concern is the lack of comparison with other existing methods applicable to the task at hand - the authors cite a number of relevant papers/algorithms from the clustering and ranking literatures, but do not compare against any of them.

A good number of real world data sets have been employed (though quite a few of the networks have an unusually small number of nodes). It would have been good to see performance on at least one large data set (with thousands of nodes, at least).

Spectral methods are known to underperform in the sparse regime; it would be good to assess the sensitivity of the two proposed methods to sparsity in the measurement graph, in particular the regime where the sampling probability is below $\log(n)/n$. Meaningful experiments in this context typically have the number of nodes at least 500-1000.

It would also be useful to place the chosen synthetic models in the context of stochastic block models; the input adj matrices in Fig 3 and 4 can also be seen as instances of stochastic block models, so it can be a bit confusing to see in the Conclusion that “the comparison could be extended to include other types of structure, including stochastic block versions”

Figure 1a would be more clear if it had on top the latent/meta-graph (a triangle with directed edges in a cycle, encoding the cluster structure, with each cluster being a meta-node in the meta-graph). And same for Figure 1b.

It would be good to see similar heatmaps as in Figure 3 & 4 (with ordered adj matrices) for some of the real data sets, as well.

Reviewer: 2

Comments to the Author(s)

The manuscript introduces a random graph based model selection framework for spectral methods that can detect the relative importance of periodic and linear hierarchies in directed networks. It is clearly and nicely written, first providing theoretical grounding for the authors' contributions, then illustrating the methods and framework on synthetic and real data. I recommend the manuscript for publication subject to minor revisions as below:

- Certain parts in the motivation section 1 would benefit from further elaboration. When the authors state "because there is a greater variety of possible structures", it would be good to add a sentence to substantiate this. When they state "This hierarchy may be periodic or linear, depending on whether there are well-defined start and end groups", I think more detail is needed to explain the concepts "periodic" and "linear" intuitively to the reader and to motivate why/when they're important in real-world applications. These concepts are quite key to the paper. The authors could then perhaps link back to this in the real world section. A minor point in figure 1 is to explicitly state that colours represent clusters in the caption.

- The generality of the model selection contribution of the paper should be made clearer. In the abstract the authors state "We develop a general framework that allows us to associate methods based on optimization formulations with maximum likelihood problems on random graphs." This sounds potentially very useful, as most methods for detecting mesoscale structure are based on optimization formulations. However, the generality of the framework doesn't quite come through later on, e.g. when the authors list their main contributions in the motivation section. Is the framework only application to spectral methods? If so, why? In any case, it would be helpful to the reader if the authors were clearer about precisely how "general" the framework is both in the motivation and the conclusion.

- In bullet point 1 of contributions, perhaps "we propose two new classes" would be more appropriate so that "both" in bullet point three can be easily linked back to that. Again in bullet point 1, it would be good to include a citation after "unusual property". More generally, the main contributions bullet points could use some polish.

- 2b: can the authors give a stronger motivation for why they specifically focus on the Magnetic Laplacian algorithm and the Trophic Laplacian algorithm? The discussion of other related techniques is rather "in passing", and there are many techniques not mentioned at all. I am not suggesting anything too detailed, but it would be good if the authors offered some convincing motivation for their focus.

- In terms of practical use, it would be good to make it clear if the methods are supervised (e.g. number of clusters assumed to be known) or not, and how well they scale with the size of the network.

- In the conclusion the authors briefly mention a potential extension to weighted networks, can more be said here and/or can the authors connect what they state to the extensions presented in the manuscript (e.g. to weights in $(0,1)$)? Can anything be said about other types of weights (e.g. integers)?

- Minor points: on p. 13 the figures loaded with some difficulty (worth double checking) and in the authors' public Git repo it would be good to include further detail in the readme file. Eg the

authors could perhaps clarify which function corresponds to which method/model/ model selection framework.

===PREPARING YOUR MANUSCRIPT===

===PREPARING YOUR REVISION IN SCHOLARONE===

- 1) One version identifying all the changes that have been made (for instance, in coloured highlight, in bold text, or tracked changes);
 - 2) A 'clean' version of the new manuscript that incorporates the changes made, but does not highlight them.
 - An individual file of each figure (EPS or print-quality PDF preferred [either format should be produced directly from original creation package], or original software format).
 - An editable file of each table (.doc, .docx, .xls, .xlsx, or .csv).
 - An editable file of all figure and table captions.
- Note: you may upload the figure, table, and caption files in a single Zip folder.
- Any electronic supplementary material (ESM).
 - If you are requesting a discretionary waiver for the article processing charge, the waiver form must be included at this step.
 - If you are providing image files for potential cover images, please upload these at this step, and inform the editorial office you have done so. You must hold the copyright to any image provided.
 - A copy of your point-by-point response to referees and Editors. This will expedite the preparation of your proof.

- Ensure that your data access statement meets the requirements at <https://royalsociety.org/journals/authors/author-guidelines/#data>. You should ensure that you cite the dataset in your reference list. If you have deposited data etc in the Dryad repository, please only include the 'For publication' link at this stage. You should remove the 'For review' link.
- If you are requesting an article processing charge waiver, you must select the relevant waiver option (if requesting a discretionary waiver, the form should have been uploaded at Step 3 'File upload' above).
- If you have uploaded ESM files, please ensure you follow the guidance at <https://royalsociety.org/journals/authors/author-guidelines/#supplementary-material> to include a suitable title and informative caption. An example of appropriate titling and captioning may be found at https://figshare.com/articles/Table_S2_from_Is_there_a_trade-off_between_peak_performance_and_performance_breadth_across_temperatures_for_aerobic_scope_in_teleost_fishes_/3843624.

Author's Response to Decision Letter for (RSOS-211144.R0)

See Appendix A.

Decision letter (RSOS-211144.R1)

Dear Miss Gong,

I am pleased to inform you that your manuscript entitled "Directed Network Laplacians and Random Graph Models" is now accepted for publication in Royal Society Open Science.

on behalf of Dr Robert MacKay (Associate Editor) and Mark Chaplain (Subject Editor)
openscience@royalsociety.org

Appendix A

Author Response:

Manuscript RSOS-211144 "Directed Network Laplacians and Random Graph Models" submitted to Royal Society Open Science

We thank the referees and editors for their feedback. We have revised the manuscript in light of their comments. Nontrivial changes to the manuscript appear in highlighted text, and we list below our point-by-point responses to the referees' feedback.

Reviewer: 1

Comments to the Author(s)

The main focus of this work is on the development of Laplacian-based methods for uncovering clustered hierarchical structure in directed networks. Compared to the undirected counterpart, the directed case has received a lot less attention in the recent and distant literature, though a number of very recent works in this area have attracted significant attention, due to their applicability to various real world settings. From this perspective, the present piece of work is very timely, and a good addition to the literature.

1. The two algorithms considered are not new, but rather direct consequences of previous works in [5] and [7]. The authors could clarify better/earlier in the paper what exactly is meant by "establish connections between members of these random graph classes and algorithms from [7] and [5]" (as in the second bullet of the contribution). Should the reader understand that, in some sense, some reverse engineering is being done, by taking two established algorithms and finding suitable probabilistic models which they do a good job solving?

We have clarified this in the revised contribution section.

2. A set of theoretical results are provided to quantify the relative strength of the two possible structures: periodic versus linear directed hierarchy, using a likelihood ratio. The authors could discuss the case when neither structure is inherent in the data, and means by which one can draw such a conclusion. Some guidance could be provided for how one might design a statistical significance test for a given uncovered structure (so not for the pairwise comparison).

The paper focuses on the well-defined question of whether the data is better explained by a linear or periodic model. We feel that the question of whether either, both or neither structure is "present" in a network is more delicate and loosely defined. One approach would be to come up with a null model, but it is not clear to us how to justify such a choice. We have therefore mentioned this issue at the end of the manuscript as a possible topic for future work.

3. My only main concern is the lack of comparison with other existing methods applicable to the task at hand - the authors cite a number of relevant papers/algorithms from the clustering and ranking literatures, but do not compare against any of them.

There are relatively few node reordering algorithms for directed networks. In response to other comments (below), we have extended our description of this literature in Section 2(b). In terms of "comparison", it is not clear to us how to introduce other algorithms into the tests if they do not have a clear objective function that leads to a random graph interpretation, and hence a likelihood.

4. A good number of real world data sets have been employed (though quite a few of the networks have an unusually small number of nodes). It would have been good to see performance on at least

one large data set (with thousands of nodes, at least). Spectral methods are known to underperform in the sparse regime; it would be good to assess the sensitivity of the two proposed methods to sparsity in the measurement graph, in particular the regime where the sampling probability is below $\log(n)/n$. Meaningful experiments in this context typically have the number of nodes at least 500-1000.

We note that two real networks in Table 1 have more than 500 nodes. Some are smaller since we extract the largest connected components to make sure the algorithm gives meaningful results. We are guessing that the comment about underperformance on sparse data refers to spectral clustering algorithms; we are not aware of results in this direction for spectral reordering.

5. It would also be useful to place the chosen synthetic models in the context of stochastic block models; the input adj matrices in Fig 3 and 4 can also be seen as instances of stochastic block models, so it can be a bit confusing to see in the Conclusion that “the comparison could be extended to include other types of structure, including stochastic block versions”

Fig 3 and 4 are generated using the range-dependent model. They look similar to a stochastic block model, but the underlying probabilities depend on distance from the diagonal and hence are not constant in rectangular blocks.

6. Figure 1a would be more clear if it had on top the latent/meta-graph (a triangle with directed edges in a cycle, encoding the cluster structure, with each cluster being a meta-node in the meta-graph). And same for Figure 1b.

Based on this remark, and a remark below, we have added text to the figure caption that explains the colouring.

7. It would be good to see similar heatmaps as in Figure 3 & 4 (with ordered adj matrices) for some of the real data sets, as well.

We have added these pictures for two of the real data sets.

Reviewer: 2

Comments to the Author(s)

The manuscript introduces a random graph based model selection framework for spectral methods that can detect the relative importance of periodic and linear hierarchies in directed networks. It is clearly and nicely written, first providing theoretical grounding for the authors' contributions, then illustrating the methods and framework on synthetic and real data. I recommend the manuscript for publication subject to minor revisions as below:

1. Certain parts in the motivation section 1 would benefit from further elaboration. When the authors state "because there is a greater variety of possible structures", it would be good to add a sentence to substantiate this. When they state "This hierarchy may be periodic or linear, depending on whether there are well-defined start and end groups", I think more detail is needed to explain the concepts "periodic" and "linear" intuitively to the reader and to motivate why/when they're important in real-world applications. These concepts are quite key to the paper. The authors could then perhaps link back to this in the real world section. A minor point in figure 1 is to explicitly state that colours represent clusters in the caption.

We have added further text in the motivation section, as suggested.

2. The generality of the model selection contribution of the paper should be made clearer. In the abstract the authors state "We develop a general framework that allows us to associate methods based on optimization formulations with maximum likelihood problems on random graphs." This sounds potentially very useful, as most methods for detecting mesoscale structure are based on optimization formulations. However, the generality of the framework doesn't quite come through later on, e.g. when the authors list their main contributions in the motivation section. Is the framework only application to spectral methods? If so, why? In any case, it would be helpful to the reader if the authors were clearer about precisely how "general" the framework is both in the motivation and the conclusion.

We have added text to clarify the scope of this framework in the abstract.

3. In bullet point 1 of contributions, perhaps "we propose two new classes" would be more appropriate so that "both" in bullet point three can be easily linked back to that. Again in bullet point 1, it would be good to include a citation after "unusual property". More generally, the main contributions bullet points could use some polish.

We have edited the bullet points. We are not aware of other models that use this property, so we have not been able to add a citation.

4. 2b: can the authors give a stronger motivation for why they specifically focus on the Magnetic Laplacian algorithm and the Trophic Laplacian algorithm? The discussion of other related techniques is rather "in passing", and there are many techniques not mentioned at all. I am not suggesting anything too detailed, but it would be good if the authors offered some convincing motivation for their focus.

These algorithms can be motivated by an optimization problem (a) that relaxes into an eigenvalue problem and (b) has a random graph/likelihood interpretation. These features make them suitable for the framework that we develop. We have added text in order to emphasize this issue in Section 2(b).

5. In terms of practical use, it would be good to make it clear if the methods are supervised (e.g. number of clusters assumed to be known) or not, and how well they scale with the size of the network.

The methods are unsupervised. The main computation required for the reordering is finding one or two dominant eigenvector/eigenvalue pairs. Assuming that the network is sparse (each node has a finite degree) and that the power method gives the required accuracy in a finite number of iterations, this is an $O(n)$ computation. Computing the corresponding likelihood is $O(n^2)$. We have added this information to the manuscript in Section 3(d).

6. In the conclusion the authors briefly mention a potential extension to weighted networks, can more be said here and/or can the authors connect what they state to the extensions presented in the manuscript (e.g. to weights in $(0,1)$)? Can anything be said about other types of weights (e.g. integers)?

We have added an extra sentence before Theorem 3.3 and 3.5 to indicate that weights coming from a discrete-valued random variable could be treated similarly.

7. Minor points: on p. 13 the figures loaded with some difficulty (worth double checking) and in the authors' public Git repo it would be good to include further detail in the readme file. Eg the authors could perhaps clarify which function corresponds to which method/model/model selection framework. We have revised the readme to the Git repo.